# Near physiological spectral selectivity of cochlear optogenetics

Alexander Dieter [1,2], Carlos J. Duque-Afonso [1,2,3], Vladan Rankovic [1,4,5], Marcus Jeschke [1,4,6,7] & Tobias Moser [1,2,3,4,7]

Cochlear implants (CIs) electrically stimulate spiral ganglion neurons (SGNs) and partially restore hearing to half a million CI users. However, wide current spread from intracochlear electrodes limits spatial selectivity (i.e. spectral resolution) of electrical CIs. Optogenetic stimulation might become an alternative, since light can be confined in space, promising artificial sound encoding with increased spectral selectivity. Here we compare spectral selectivity of optogenetic, electric, and acoustic stimulation by multi-channel recordings in the inferior colliculus (IC) of gerbils. When projecting light onto tonotopically distinct SGNs, we observe corresponding tonotopically ordered IC activity. An activity-based comparison reveals that spectral selectivity of optogenetic stimulation is indistinguishable from acoustic stimulation for modest intensities. Moreover, optogenetic stimulation outperforms bipolar electric stimulation at medium and high intensities and monopolar electric stimulation at all intensities. In conclusion, we demonstrate better spectral selectivity of optogenetic over electric SGN stimulation, suggesting the potential for improved hearing restoration by optical CIs.

[1] Institute for Auditory Neuroscience and InnerEarLab, University Medical Center Göttingen, 37075 Göttingen, Germany. [2] Göttingen Graduate School for Neurosciences and Molecular Biosciences, University of Göttingen, 37075 Göttingen, Germany. [3] Auditory Neuroscience Group, Max Planck Institute for Experimental Medicine, 37075 Göttingen, Germany. [4] Auditory Neuroscience and Optogenetics Laboratory, German Primate Center, 37077 Göttingen, Germany. [5] Restorative Cochlear Genomics Group, Auditory Neuroscience and Optogenetics Laboratory, German Primate Center, 37077 Göttingen, Germany. [6] Cognitive Hearing in Primates Group, Auditory Neuroscience and Optogenetics Laboratory, German Primate Center, 37077 Göttingen, Germany. [7] These authors jointly supervised this work: Marcus Jeschke, Tobias Moser. Correspondence and requests for materials should be addressed to T.M. (email: tmoser@gwdg.de)

By stimulating spiral ganglion neurons (SGNs) electrically, cochlear implants (eCIs) provide the auditory system of profoundly hearing impaired and deaf with information on the surrounding acoustic scene[1,2]. eCIs are considered the most successful neuroprosthesis and enable open speech comprehension in the majority of ~500,000 users. Still, there is an unmet need for improvement: Wide spread of electric current from each electrode contact activates large subsets of SGNs, limiting the number of independent stimulation channels in eCIs to <10[3–5]. This major drawback of electric stimulation restricts the amount of spectral information that CIs can provide to the user, ultimately resulting in limited perception of acoustic signals, such as speech, especially in noisy environments[5,6].

Optical stimulation of SGNs represents a novel approach to overcome this limitation of eCIs: Light can be better confined in space and, hence, optical CIs (oCIs) could activate SGNs along the tonotopic axis of the cochlea with higher spatial selectivity. This promises improved spectral resolution of artificial sound coding and consequently an increased number of independent stimulation channels[7–10]. Studies of cochlear activation using infrared stimulation have indicated that spatial (and thus spectral) spread of SGN excitation in the cochlea is small for optical stimulation, comparable to pure tone acoustic stimulation[7], while monopolar electrical stimulation led to spectrally broader SGN activation than infrared stimulation in a different study[11]. However, the energy requirement per pulse is very high for infrared stimulation (16–160 μJ[12]), the exact mechanism of neural activation is still under debate and activation of the auditory pathway could not be verified in several studies on animal models of sensorineural hearing loss[13–15].

In contrast, optogenetic stimulation of SGNs expressing Channelrhodopsins (ChRs[16,17]) enables neural excitation by a well understood mechanism at lower light intensities. Indeed, stimulation of the auditory system using fiber-based oCI has greatly advanced in the past years: a proof of principle study employing cochlear optogenetics in transgenic mice demonstrated optical activation of the auditory pathway up to the inferior colliculus (IC), where current source density analysis indicated a smaller spread of SGN excitation for optical than for monopolar electrical stimulation[8]. Subsequent studies on mice in which SGNs were virally transduced during the first postnatal week using the fast-gating ChRs Chronos and f-Chrimson demonstrated high temporal fidelity of neural control up to several hundred Hertz by recording optogenetically driven auditory brainstem responses (oABRs) and spiking activity of individual SGNs in hearing and deaf animals[18,19]. Finally, viral transduction of SGNs in adult Mongolian gerbils was recently established and optogenetic stimulation of the auditory nerve was studied by recordings of oABRs and individual SGNs[20]. Furthermore, stimulus perception upon optogenetic stimulation of the auditory nerve was demonstrated by single unit recordings from primary auditory cortex and behavioral experiments, the latter also involving deafened animals.

However, despite the recent progress, a precise estimation of the spectral selectivity of optogenetic SGN stimulation and a rigorous comparison to acoustic and electrical stimulation is still lacking. Here, we perform multi-channel recordings of neuronal clusters (multi-unit activity (MUA)) in the tonotopically organized central nucleus of the IC (ICC) in Mongolian gerbils while stimulating SGNs optogenetically, electrically, or acoustically. We demonstrate spatially selective optical activation of the auditory system in a tonotopic manner with a spectrally more confined SGN excitation than the one found upon monopolar and bipolar electrical stimulation. This indicates increased spectral resolution of artificial sound encoding when using optogenetic instead of electrical stimulation—and thus suggests that oCIs might overcome the major bottleneck of eCIs.

## Results

**Mapping acoustic response properties in the central nucleus of the IC.** To characterize spectral response properties of the auditory system to optogenetic stimulation of SGNs we performed electrophysiological recordings of MUA in the ICC in isoflurane-anesthetized gerbils. The ICC was chosen because of its well-defined tonotopic organization, enabling estimation of the cochlear spread of excitation of acoustic, electrical, and optogenetic stimulation. We used multiple laser-coupled optical fibers placed at three different positions along the tonotopic axis of the cochlea and recorded activity in the ICC using linear 32-channel silicon probes (Fig. 1a–c).

After placing the silicon probe, frequency response areas were constructed for each recording site using acoustic stimulation. We then derived the characteristic frequencies of the electrodes (CFs) (Fig. 1d, e) and calculated tonotopic slopes for each animal by linearly fitting the CFs as a function of recording depth (Fig. 1e, Supplementary Fig. 1). The median tonotopic slope amounted to 4.58 octaves/mm (±0.69 median average deviation, $n = 46$; Supplementary Fig. 1, inset) and did not differ between animals that underwent cochlear surgeries and naïve animals (4.61 ± 0.70 octaves/mm, $n = 38$ vs. 4.58 ± 0.60 octaves/mm, $n = 8$; two-sample $t$-test: $p = 0.69$). However, in animals that underwent cochlear surgeries, thresholds of acoustically driven multi-units in the ICC increased by 17.8 dB on average (30.3 ± 13.4 dB SPL standard deviation ($n = 246$) in naïve animals, 48.1 ± 11.6 dB SPL ($n = 208$) in animals that underwent surgery, $p < 0.001*10^{-35}$, two-sample $t$-test; Supplementary Fig. 2).

**Artificial stimulation of SGNs.** To prove stimulation of SGNs via oCI or eCI and determine time windows for potential responses, peri-stimulus time histograms (PSTHs) were constructed in response to the strongest optical and electrical stimuli (Supplementary Fig. 3). Potential responses to various stimulus intensities were then evaluated during these time windows, as well as during the presentation of tone bursts for acoustical stimulation. The majority of acoustically driven multi-units (98.1%) and all optically and electrically driven units showed a monotonic firing pattern, i.e. a rise in stimulus intensity led to a rise in firing rate (monotonicity index > 0.5; Fig. 2a–h). This observation allowed for estimation of the dynamic range (DR) based on stimulus-response functions both per individual multi-unit and per animal with a similar procedure. The DR was calculated as the range of stimulus intensities that led to a monotonic increase in firing rates in the range from 10% above baseline activity (average response to the three lowest stimulus intensities) to 10% below maximum response (averaged response to the three highest stimulus intensities). We note that saturation was not achieved for most of the optogenetically driven multi-unit responses (Fig. 2f), such that the apparent DR underestimates the true DR for optogenetic stimulation.

The average apparent DR of individual multi-units in response to optical stimulation amounted to 7.8 dB (mW; ±3.7 s.d.; $n = 762$), which exceeded the DR in response to monopolar electrical stimulation (6.9 ± 4.2 dB [μA], $n = 1515$, $p < 0.05$; one-way ANOVA and post-hoc pairwise comparison) and was comparable to the DR in response to bipolar electrical stimulation (8.3 ± 5.1 dB [μA], $n = 572$; $p = 0.65$; Fig. 2h). The DR in response to acoustic stimulation exceeded all modalities of artificial cochlear stimulation and amounted to 21.4 ± 10.1 dB (SPL) ($n = 1531$; $p < 0.001$). The grand average DR (derived from the mean of the averaged multi-units per animal) was similar between responses

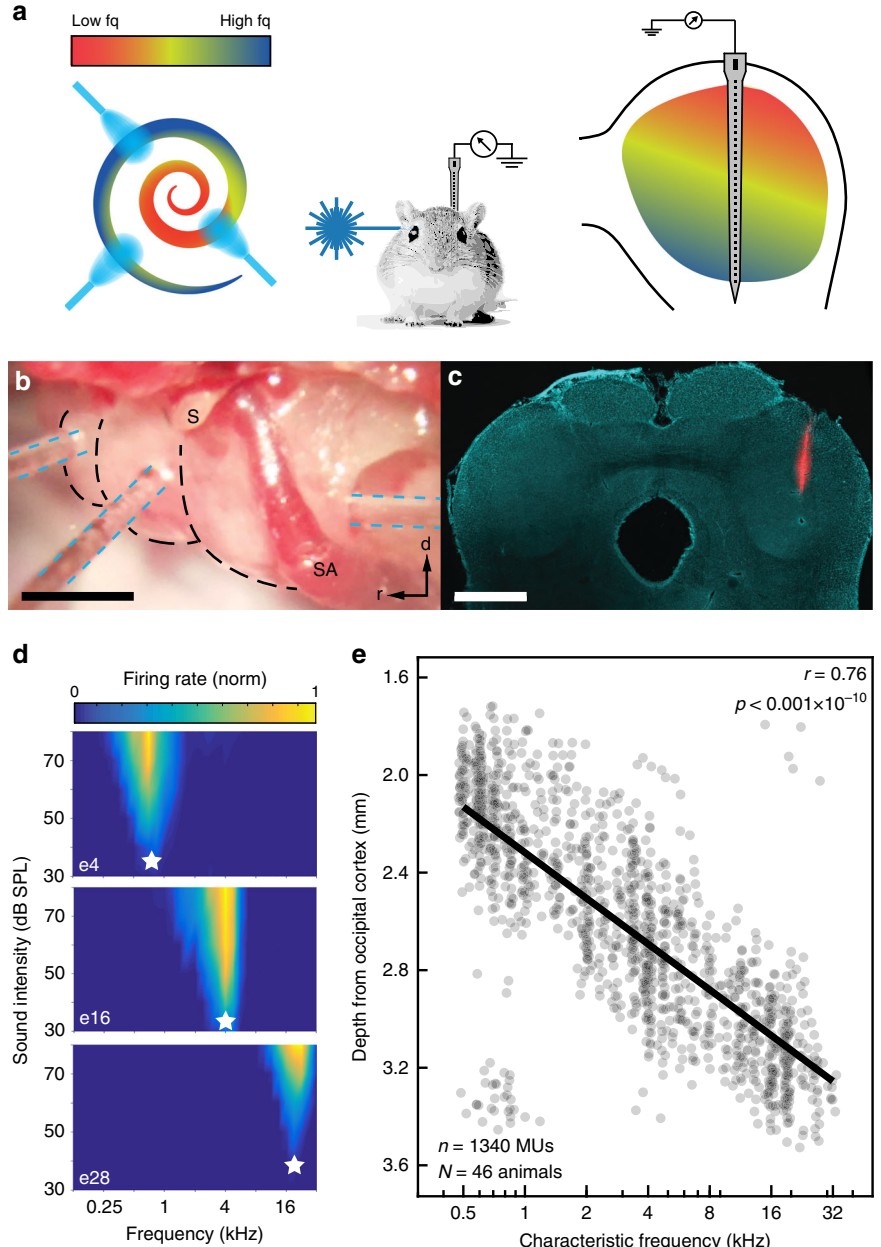

**Fig. 1** Experimental layout and acoustic response properties. **a** Experimental design. **b** Optical fibers (blue dashed lines) inserted via cochleostomies at the apical and mid-turn of the cochlea (black dashed lines), as well as in the round window (from left to right). S: stapes, SA: stapedial artery. Scale bar: ~1 mm. **c** DiI-stained electrode track (red) in a DAPI-stained (cyan) coronal section of the inferior colliculus. Scale bar: 1 mm. **d** Frequency response areas and characteristic frequencies (CFs; white stars) recorded at electrode 4, 16, and 24 (e4, 16, 28) in one animal. **e** CFs as a function of recording depth. Solid line: linear fit of all CFs, according to Pearson's correlation coefficient. Data is pooled from all animals (virus-injected as well as non-injected). Source data of (e) is provided as a source data file

to optical (10.7 ± 3.4 dB standard deviation, $n = 34$), monopolar (10.7 ± 3.5 dB, $n = 48$; $p = 0.99$), and bipolar electrical stimulation (12.2 ± 3.9 dB, $n = 18$; $p = 0.88$), while the DR of acoustic stimulation amounted to 32.3 ± 10 dB ($n = 73$; Fig. 2j; one-way ANOVA and post-hoc pairwise comparison). Comparison of the maximal strength of cochlear excitation—measured in $d'$ values based on evoked firing rates—revealed that optical stimulation could drive neurons in the ICC as effectively as electrical stimulation (max. response: 4.4 ± 1.0 $d'$ values mean and SD, $n = 34$; compared to 4.5 ± 0.7 $d'$ values for monopolar stimulation ($n = 48$; $p = 0.99$) and 4.9 ± 0.6 $d'$ values for bipolar stimulation ($n = 18$; $p = 0.79$), respectively), whereas acoustic stimulation of non-injected animals yielded stronger activation (9.0 ± 2.9 $d'$-

values; one-way ANOVA and post-hoc pairwise comparison; Fig. 2k).

We note that no responses to optical stimulation of SGNs were observed in non-injected control animals, excluding excitation unrelated to the optogenetic mechanism, such as opto-acoustic or opto-thermal effects and thus proving the specificity of optogenetic SGN stimulation (Supplementary Fig. 4).

**Spectral spread of fiber-based cochlear optogenetics**. To thoroughly characterize the spread of excitation upon optical stimulation, we used a multi-site approach to project light onto SGNs at three distinct tonotopic positions. For this purpose, optical

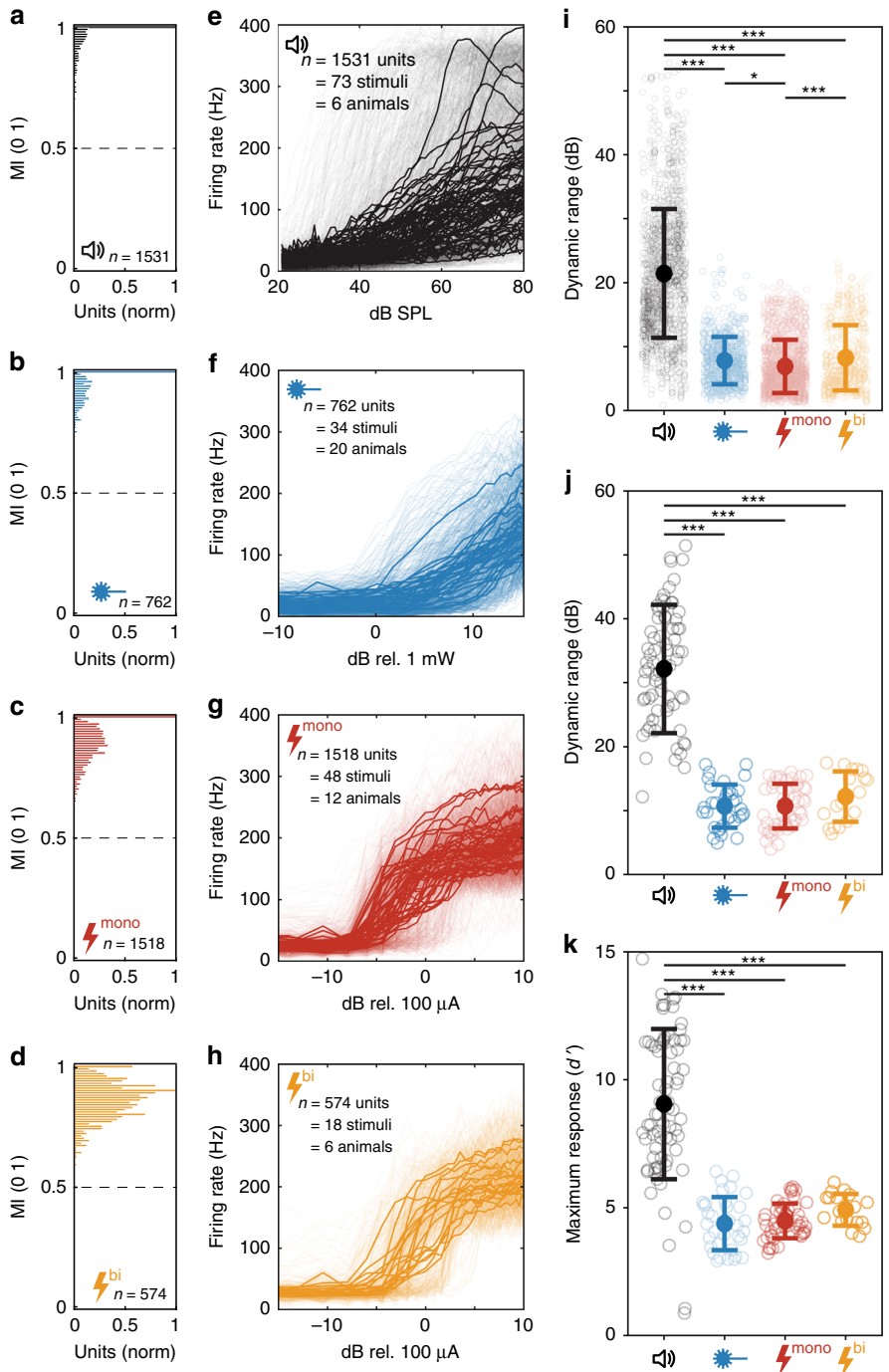

**Fig. 2** Acoustic, optogenetic, and electric activation of the auditory system. **a–d** Monotonicity indices (MI) normalized to the maximum bin for acoustic stimulation in non-injected animals (**a**), optogenetic stimulation in virus-injected animals (**b**), monopolar (**c**), and bipolar electric stimulation in non-injected animals (**d**). Dashed lines mark a monotonicity index of 0.5, above which units are considered as monotonic. **e–h** Firing rates of multi-units as a function of stimulus intensity (thin, transparent lines) and average per animal and stimulus (solid lines) for acoustic (**e**), optogenetic (**f**), monopolar (**g**), and bipolar (**h**) electric stimulation. **i** Distribution of dynamic ranges (DRs) per multi-unit in dB (SPL), dB (mW), and dB (μA). **j** DRs of averaged multi-units per animal and stimulus, units as in (**e**). **k** Maximum strength of response that could be evoked by any stimulus modality. Data in (**i**), (**j**), and (**k**) is displayed as mean ± s.d. Stars indicate statistical significance (*$p < 0.05$, **$p < 0.01$, ***$p < 0.001$), based on one-way ANOVA and post-hoc pairwise comparison. Only significant differences are indicated. Source data of all panels is provided as a source data file

fibers were inserted into the cochlea (i) via the round window (high-frequency base) and via cochleostomies in the (ii) middle (mid-frequency), and (iii) apical cochlear turn (low-frequency) (Fig. 1b).

For comparing the optogenetic spread of excitation to that of acoustic and electrical stimulation, we also performed recordings while stimulating naïve animals that did not undergo any surgical manipulation of the cochlea using pure tones and stimulating non-injected animals with a four-channel eCI using monopolar electrical stimulation, where the return electrode was placed outside of the bulla tympanica and bipolar electrical stimulation, where the electrode next to the stimulation electrode (in basal

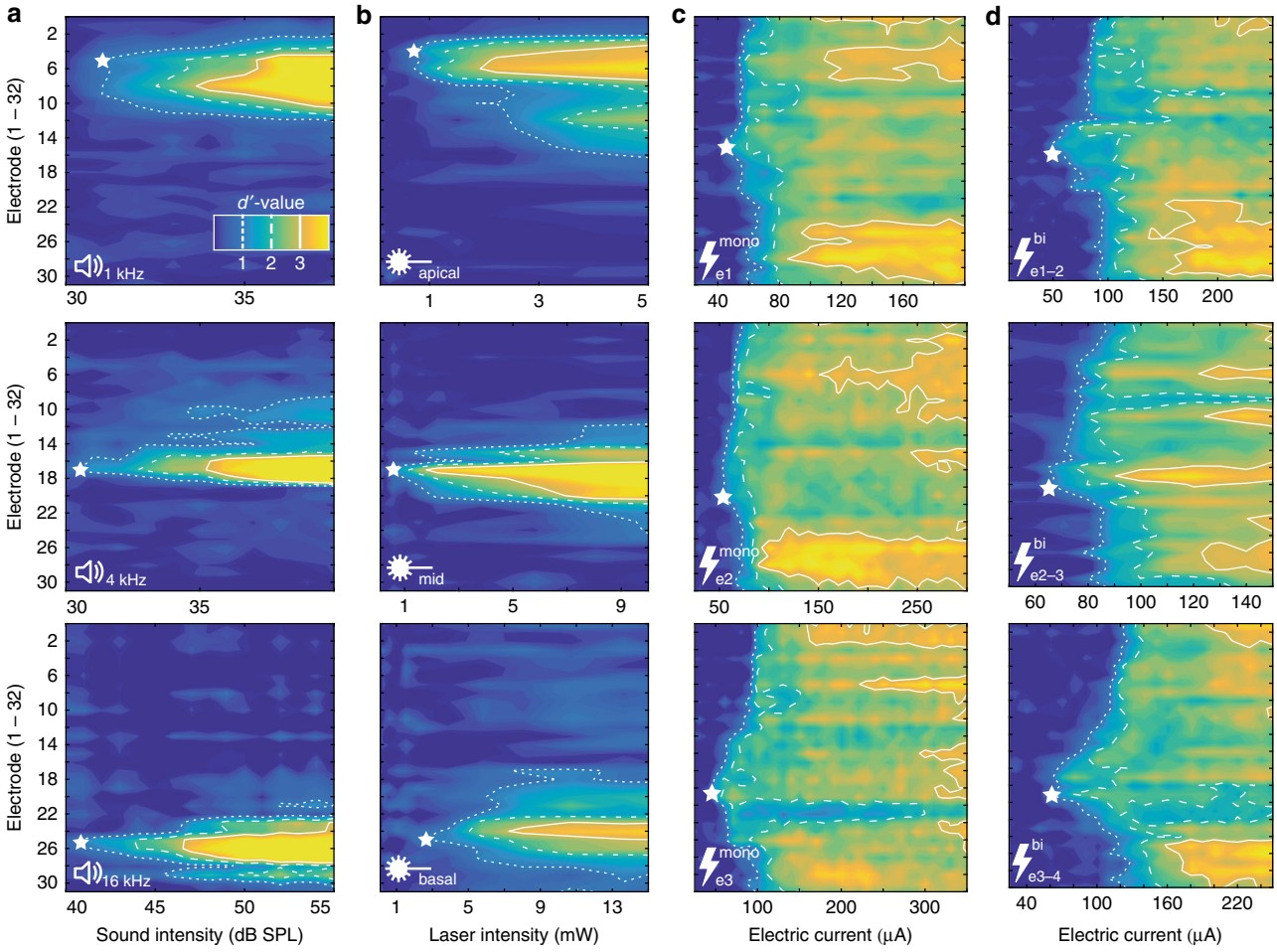

**Fig. 3** Exemplary spatial tuning curves (STC) for different stimulus modalities. **a** STCs in response to pure tones of different frequencies in naive animals. **b** STCs in response to laser pulses delivered via optical fibers placed at different positions in the cochlea of virus-injected animals. **c** STCs in response to electric current delivered via different electrodes (e1, e2, and e3) of an electrical cochlear implant in non-injected animals in the monopolar configuration. **d** STCs in response to electric current delivered via the same electrodes in bipolar configuration. White stars indicate the best electrode (BE) of each STC. The color scale in (**a**) applies to all panels displayed in this figure

direction) served as the return electrode. Neuronal activity was recorded in response to a given modality of varying stimulation intensity (pure tones, optically via one of the fibers, or electrical stimulation via one of the four eCI electrodes (Fig. 3)).

To characterize the change in multi-unit firing rates to changing stimulation levels across different modalities we employed a method based on signal detection theory[21,22]. A cumulative $d'$-value based on multi-unit spike rates in response to increasing stimulation intensities was calculated, starting in the absence of stimulation (i.e. zero intensity). Cumulative $d'$-values for increasing stimulus intensities were then sorted into a response matrix according to the electrode they were recorded from and iso-$d'$-contour-lines were drawn at integer $d'$-values in order to construct spatial tuning curves (STC; Figs. 3 and 4). As in previous studies on eCI, a $d'$-value of 1 was defined as the threshold and the recording electrode with the lowest threshold was defined as the best electrode (BE;[21,22]). Average thresholds amounted to 2.67 mW for optical, and 45.8/52.8 µA for monopolar/bipolar electrical stimulation (Supplementary Fig. 5).

In order to quantify the cochlear spread of excitation, we measured the distance between all active electrodes ($d' > 1$) at the stimulus intensity at which the BE reached a given $d'$ value (i.e. 1.5, 2, 2.5, or 3; Fig. 4a). In some cases, more than one peak (defined as electrodes below threshold separating electrodes above threshold) has been observed for each stimulus modality

(acoustic: 33/304; optogenetic: 27/101; monopolar electric: 25/192; bipolar electric: 7/72). In these cases, the dorsal-most and ventral-most electrodes with a significant response ($d' > 1$) have been considered as the boundaries of the STC to avoid underestimation of the spread of excitation (Supplementary Fig. 6). Using the tonotopic slopes calculated above, spatial spread in each animal (measured in distance between the electrodes) could then be translated into spectral spread of cochlear excitation (measured in octaves). Measuring at fixed significance of response strengths—i.e. at identical levels of activation—rather than at fixed stimulus intensities, the estimation of the spread of excitation becomes independent of the stimulus' nature and makes neural activation by different modalities more comparable.

Plotting the focus of activation (i.e. BE) as a function of stimulation frequency visualizes the natural tonotopic axis of the IC with an average tonotopic slope of 4.48 octaves/mm (Pearson's correlation coefficient $r = 0.80$, $p < 0.001$; Fig. 3a, also see Figs. 1 and 4b). Similarly, neuronal responses could be shifted systematically from the dorsal to the ventral IC when stimulating optically at the apex, mid or base of the cochlea in AAV-injected animals ($p < 0.001$; Figs. 3b, 4c, Supplementary Fig. 7a). A systematic shift of activation was also apparent for eCI stimulation via different electrodes upon monopolar electrical stimulation ($p < 0.001$; Fig. 3c, also see Fig. 4d) but was not significant when stimulating in bipolar configuration ($p = 0.18$).

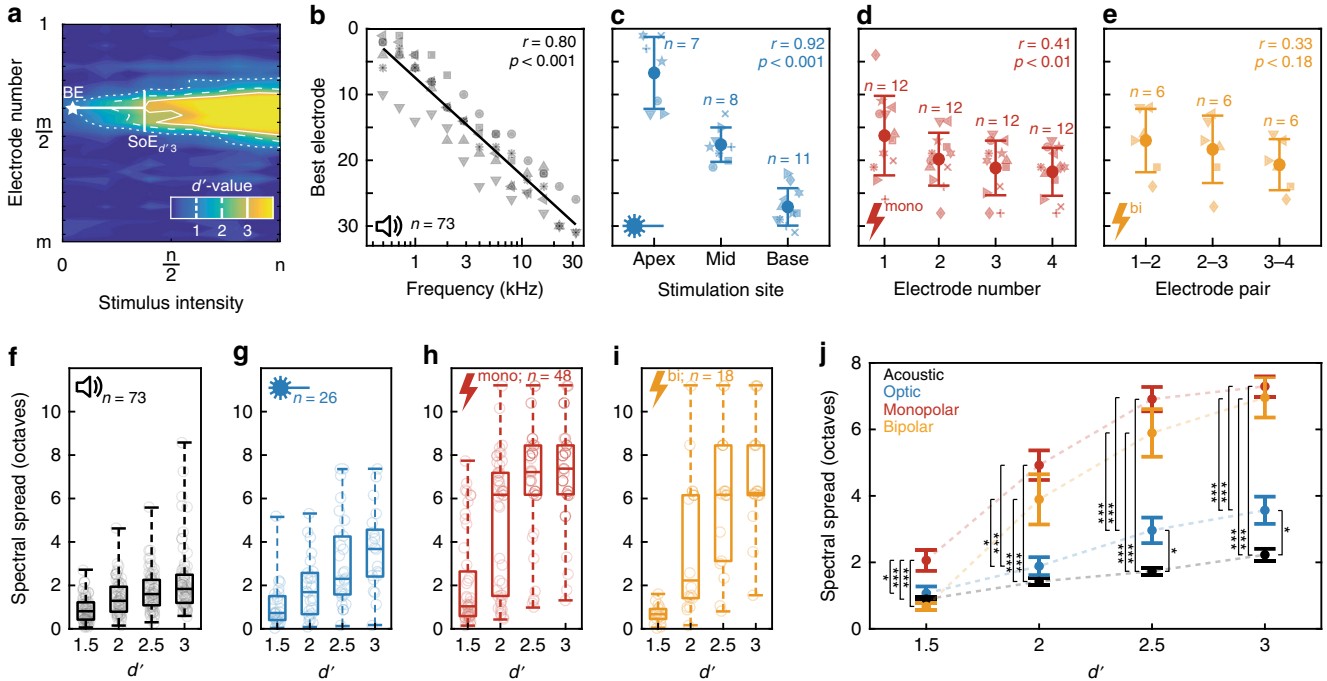

**Fig. 4** Quantification of spatial tuning curves (STCs). **a** Quantification of STCs: STC in response to a 2 kHz pure tone in a non-injected gerbil. Cumulative $d'$ values are color-coded in a matrix sorted according to electrode number (ordinate) and stimulus intensity (abscissa). The electrode with the lowest threshold ($d' = 1$) is defined as the best electrode (BE, white star) and the spread of excitation (SoE) is measured at different activation strengths (exemplary shown for a $d'$ of 3 at the BE). **b** BEs as a function of stimulation frequency recorded from non-injected gerbils. Solid line: linear fit of tonotopic slope. Different symbols mark different animals. **c** BEs as a function of optical stimulation site in virus-injected animals, including mean and standard deviation for each stimulation site. Symbols mark different animals. **d** BEs as a function of stimulation electrode for monopolar electrical stimulation in non-injected animals, including mean and standard deviation for each electrode. **e** BEs as a function of stimulating electrode pair for bipolar electrical stimulation in non-injected animals, including mean and standard deviation for each electrode pair. Pearson's correlation coefficient $r$ and the corresponding $p$-values were calculated for panel **b**–**e**. **f**–**i** Spectral spread has been quantified at different $d'$-values for acoustic (**f**), optogenetic (**g**), monopolar (**h**) and bipolar electric stimulation (**i**). Box plots indicate minimum (lower) and maximum (upper) whisker, median (center line), as well as 25th percentile and 75th percentile. **j** Mean and SEM for the spread of excitation upon acoustic, optogenetic, monopolar, and bipolar electric stimulation. Stars indicate statistical significance (*$p < 0.05$, **$p < 0.01$, ***$p < 0.001$), according to repeated-measures ANOVA and post-hoc pairwise comparison tests. Only significant differences have been indicated. Source data of panels **b**–**j** is provided as a source data file

Nonetheless, when probing tonotopy of stimulation by correlating stimulating electrode (pairs) with the CF recorded at the BE it was significant for both monopolar and bipolar electrical stimulation ($r = 0.31$, $p < 0.05$ and $r = 0.6$, $p < 0.01$, respectively; Supplementary Fig. 7b, c). Note that the layout of the eCI might have contributed to the less pronounced tonotopic activation, as the four electrodes only covered 1.8 mm in the case of monopolar and 1.2 mm in the case of bipolar electrical stimulation and thus were not distributed along the whole cochlea, whereas optical fibers have been placed in a way to cover large parts of the cochlear spiral.

In order to better understand the spread of light upon optical stimulation and the variance introduced by fiber-positioning, we modeled ~3,000,000 optical rays in a Monte-Carlo simulation where optical fibers were placed at the corresponding positions within a model of the gerbil cochlea reconstructed from x-ray phase-contrast tomography (Supplementary Fig. 8). Peak illumination of the spiral ganglion in Rosenthal's canal was observed at tonotopic places corresponding to 1.01, 6.9, and 22.89 kHz, spanning a total of 4.43 octaves. Varying the angle of light projection shifted the peak of illumination at the spiral ganglion, which (just as precise axial positioning of the fiber aperture) contributes to the variance of the experimentally observed BEs and CFs upon optical stimulation.

Spread of excitation at increasing $d'$ values grew for all stimulus modalities (Fig. 4f–i). At a $d'$ of 1.5 and 2, the spread of excitation

of optogenetic stimulation was indistinguishable from that upon pure tones suggesting that near physiological frequency resolution might be achievable in future oCI for modest stimulation strength (repeated-measures ANOVA and post-hoc pairwise comparison tests; Fig. 4j). At higher activation levels ($d'$ of 2.5 and 3) the spread of excitation was significantly higher for optogenetic stimulation ($p < 0.05$) as compared to acoustic stimulation. However, optogenetic stimulation outperformed monopolar eCI stimulation at all activation strengths ($p < 0.05$ at a $d'$ of 1.5; $p < 0.001$ at all other activation strengths). Furthermore, optogenetic stimulation performed significantly better than bipolar electrical stimulation at medium ($d'$ of 2; $p < 0.05$) and high activation levels ($d'$ of 2.5/3; $p < 0.001$), while no difference was found at low activation levels ($d'$ of 1.5; $p = 0.77$). These findings were identical when the spectral spread of excitation was not normalized by the tonotopic axis of each animal, i.e. when measuring the spread of excitation in terms of spatial activation in the IC (Supplementary Fig. 9). For optical stimulation via the round window we also compared the fiber orientation projecting light onto SGNs of (i) the high-frequency base (used in the present comparison (Figs. 3 and 4) and (ii) the orientation pointing towards the cochlear apex and thus projecting light along the modiolar axis (used in our previous study[20]), which led to a rather broad activation of the spiral ganglion (Supplementary Fig. 10). Thus, the precise projection of light to the spiral ganglion is of critical importance to achieve high spectral resolution by oCI stimulation.

These findings suggest that optical excitation of SGNs happens in a more spatially confined manner than electrical stimulation and hence can achieve better spectral resolution, provided appropriate projection of the light.

## Discussion

In this study, we scrutinized the spectral spread of excitation for optogenetic stimulation of SGNs and compared it to physiological hearing and hearing with eCI employing monopolar stimulation, used in the majority of clinical eCIs[23], as well as bipolar stimulation. We could demonstrate a major advantage of optogenetic over monopolar and bipolar electrical stimulation for this highly relevant parameter of artificial sound encoding. In fact, the study indicates that optogenetic stimulation can achieve near physiological frequency selectivity at low to modest levels of activation. We attribute this primarily to the spatial confinement of optical stimulation even with the rather crude oCI implementation in our current study.

A prerequisite for quantifying cochlear spread of excitation based on ICC measurements is reliable and reproducible positioning of electrode arrays along the tonotopic axis of the ICC. Hence, we placed the array under guidance of neuronal responses to acoustic stimuli. The expected correlation of electrode CF and depth in the ICC was observed for each animal in this study. The median tonotopic slope amounted to 4.58 octaves/mm based on CFs and to 4.48 octaves/mm based on the BEs, which compares well to literature (4.08 and 4.37 octaves/mm, respectively[24,25]). A few units in ventral IC regions were found to be responsive to low-frequency tones, indicating that recording sites for these units were likely outside the ICC[24]. However, since these were only 30 out of 1340 units, 97.76% of multi-units recorded in this study can be considered to primarily originate from the ICC. As the external cortex, covering the ICC dorsally and laterally, is thin (<150 μm[26,27]), we have not attempted to separate the expectedly few cortical neurons contributing to the data set.

Optogenetic stimulation of SGNs at different positions of the cochlear spiral evoked neural activity in tonotopically corresponding ICC regions: apical stimulation evoked activity in the dorsal, low-frequency regions of the ICC while baso-cochlear stimulation excited ventral, high-frequency parts. Tonotopic ICC activation was less obvious when using electrical stimulation. We note that appropriate positioning of the oCI/eCI within the cochlea is critical for its tonotopic activation: Optical fibers were placed in three spatially distinct positions to cover a large tonotopic range (see Figs. 1b, 4c, Supplementary Fig. 5a; i.e. regions coding centered around 0.6, 3.6, and 13.3 kHz from apical, mid, and basal stimulation, spanning a total of 4.5 octaves, i.e. 56% of the gerbils hearing range). This was confirmed by modeling optical rays from the fiber aperture placed in a reconstructed gerbil cochlea (Supplementary Fig. 8). In contrast, electrodes of the eCI spanned 1.8 mm (monopolar) and 1.2 mm (bipolar) of the scala tympani, covering only 16.4% and 10.9% (i.e. 0.29 and 0.19-fold of optical stimulation) of the cochlear length[28]. Experimentally, average BE IDs in response to optical stimulation upon apical and basal stimulation were 6.7 and 27.9, whereas in response to electrical stimulation the electrode IDs for apical-most and basal-most monopolar or bipolar stimulation were 16.5/ 21.8 or 17/20.7 (i.e. 0.25-fold of 0.17-fold optical stimulation, respectively). These ICC estimates agree well with cochlear tonotopic ranges estimated above for optical and electrical stimulation, suggesting that tonotopic activation was achieved in a comparable manner.

The spread of cochlear excitation determines the spectral resolution of artificial sound coding which is limited with current eCIs. A previous study on cats (where the spread of excitation was quantified by measuring the width of IC activation 6 dB above threshold) reported 2.55 and 4.94 octaves of neural activation for bipolar and monopolar stimulation, compared to 0.6 octaves for pure tones[21]. Electrical stimulation via an intraneural electrode array (penetrating the auditory nerve) outperformed eCIs conventionally placed in the scala tympani, activating only 1.4 octaves in the same study. When analyzing our data as done in this study (6 dB above threshold, corresponding to a mean/SD $d'$ of 3.02 ± 0.64), the spread of excitation in our study amounted to 1.81 ± 0.7 (SD; $n = 71$), 2.91 ± 1.6 ($n = 20$), 6.36 ± 2.3 ($n = 44$), and 6.07 ± 2.6 ($n = 16$) octaves for acoustic, optogenetic, monopolar, and bipolar electric stimulation, respectively (Supplementary Fig. 11).

Since the species used in these studies differ both in physiology and anatomy, we suggest normalizing the spread of excitation to the corresponding spread of excitation upon pure tone stimulation employed in each study to facilitate a better comparison. By doing so, the spread of excitation with the penetrating array of the above-mentioned study corresponds to 2.33-fold of the acoustic one, while bipolar and monopolar stimulation of eCI in the scala tympani amounted to 4.16-fold and 8.23-fold spread. The spread of excitation evoked by optical stimuli in our study amounted to 1.61-fold, while the ones of monopolar and bipolar electrical stimulation amounted to 3.51-fold and 3.35-fold spread, respectively. Another study, performed in guinea pigs reported 3.9-fold and 1.8-fold spread of excitation upon monopolar and bipolar electrical stimulation, respectively (measured 6 dB above threshold; compared to acoustic stimulation 20 dB above threshold)[22]. Comparing our data in this way, we found 0.74-fold spread of excitation upon optogenetic and 1.61-fold/1.53-fold excitation upon monopolar and bipolar electrical stimulation, respectively (Supplementary Fig. 11).

Not finding the previously reported advantage of bipolar electrical stimulation over monopolar stimulation in our study is most likely explained by the fact that the majority of STCs upon electrical stimulation have been underestimated. The boundaries of recorded STCs exceeded the limits of the electrode array in many cases (i.e. neural activity was evoked at all recording sites). Since the supposedly more focused bipolar stimulation already evokes activity in most parts of the cochlea (and thus the IC), less selective stimulation in the monopolar configuration could not be demonstrated. Therefore, we found bipolar stimulation to be more selective than monopolar stimulation only for weak stimuli ($d'$ of 1.5). We attribute this finding primarily to the model system we used: since the gerbil cochlea is roughly 2.5 times smaller than the cat cochlea and ~1.6 times smaller than the guinea pig cochlea, a physically similar current spread from the eCI electrode will lead to activation of a larger SGN population in the gerbil than in the cat and guinea pig[21,28–31]. Thus, the advantage of bipolar, and importantly, even more so of optical stimulation, over monopolar electrical stimulation—especially at high activation strengths—is expected to be greater in species with larger cochleae, e.g. cats or even humans (even in the guinea pig, 60% and 93% of the STCs upon monopolar and bipolar electrical stimulation exceeded the boundaries of the recording electrode array and thus might have been underestimated[22]). Despite the limited size of the gerbil cochlea, however, we could still demonstrate spatially selective SGN activation upon optogenetic stimulation, approaching the frequency resolution of natural acoustic stimulation.

Taken together, these results suggest that optical stimulation of cochlear neurons is indeed more confined in space than electrical stimulation, despite the non-optimal projection of light from a fiber aperture placed at an opening of the cochlear capsule (Supplementary Fig. 8). We suppose that the spread of excitation upon optogenetic SGN stimulation might be even lower when placing light sources into the scala tympani (i.e. closer to the

target tissue), using emitters with a lower numerical aperture or by combining light emitters with focusing lenses. Indeed, ray modeling studies indicate narrow tonotopic ranges of activation under these conditions (Supplementary Fig. 8c, d).

The output DR of individual multi-units in response to sound (21.4 dB) and electrical stimulation (6.9 and 8.3 dB for monopolar and bipolar stimulations) are in good agreement with literature values: an average DR of 20 dB per single unit in response to sound has been reported for the gerbil IC and DRs between 6.7 and 7.6 dB have been reported upon stimulation with different kinds of eCIs in the rat IC[31–33]. DR comparison of optogenetic to electrical stimulation is confounded by the fact that we did not find saturation for most multi-units at the light intensities amenable to our setup (Fig. 2e, Supplementary Fig. 12).

The resulting apparent DR of 7.8 dB (mW) was only slightly larger than the DR in response to monopolar, and comparable to bipolar electrical stimulation. We note that our DR estimation for optical stimulation refers to dB (mW), i.e. $\mathrm{DR} = 10 \times \log_{10} \frac{\mathrm{power}(90\%)}{\mathrm{power}(10\%)}$, while that of electrical stimulation was based on current amplitudes, i.e. $\mathrm{DR} = 20 \times \log_{10} \frac{\mathrm{amplitude}(90\%)}{\mathrm{amplitude}(10\%)}$. Future studies involving behavioral analysis should provide psychophysical estimates of DR, as well as of the intensity discrimination. Furthermore, optimized opsins conferring increased light sensitivity might lower the threshold of SGN activation and thus increase the DR at the lower end. Biosafety studies of long-term exposure to light illumination need to be done in order to determine safe margins for light stimulation that might limit the DR at the upper end.

An obvious limitation of our study is the use of multiple laser-coupled optical fibers placed near the cochlear lateral wall at variable distance to SGNs (Supplementary Fig. 8). This approach is not feasible for clinical translation. Due to the spatial flexibility of placing optical fibers at arbitrary cochlear positions, it was possible to access the tonotopic axis of the spiral ganglion throughout the whole cochlea, from the round window up to the apex. Even though the technical feasibility of oCIs, e.g. in the form of miniaturized LEDs on a flexible substrate[34], has been demonstrated, chronic translational experiments requiring stable multi-channel oCIs have not yet been reported to our knowledge. A second — apparent — limitation of the current study is that artificial stimulation of SGNs was done in the presence of inner hair cells, i.e. in normal hearing animals. We can assume that the presented results would not differ in deaf animals, since we have shown in a previous study that opsin expression is absent in inner hair cells due to the choice of the promotor (human synapsin) and furthermore, optogenetic excitation of SGNs is feasible in a model of sensorineural hearing loss[20]. Also, we showed that there was no effect of optical stimulation in non-injected animals, ruling out the contribution of non-optogenetic neural excitation by light.

## Methods

**Animals**. Data was recorded from 46 adult (>8 weeks of age) Mongolian gerbils (*Meriones unguiculatus*) of either sex. For each surgery, gerbils were anesthetized with Isoflurane (4% at 1 l/min for induction, 1–2% at 0.4 l/min for maintenance) and appropriate analgesia was achieved by subcutaneous injection of Buprenorphine (0.1 mg/kg BW) 30 min prior to surgery. Depth of anesthesia was monitored regularly by the absence of reflexes (hind limb withdrawal) and adjusted accordingly. During all experiments, animals were placed on a heating pad and body temperature was maintained at 37 °C. All experimental procedures were done in compliance with the German national animal care guidelines and approved by the local animal welfare committee of the University Medical Center Göttingen, as well as the animal welfare office of the state of Lower Saxony, Germany (LAVES).

**Virus purification**. Adeno-associated viruses (AAVs) were generated in HEK-293T cells (ATCC) using polyethylenimine transfection (25,000 MW, Polysciences, USA)[35,36]. In brief, triple transfection of HEK-293T cells was performed using pHelper plasmid (TaKaRa/Clontech), trans-plasmid providing viral capsid PHP.B (generous gift from Ben Deverman and Viviana Gradinaru, Caltech, USA) and cis-

plasmid containing gene of interest flanked by two ITRs in the ends. The cell line was regularly tested for mycoplasma. We harvested viral particles 72 h after transfection from the medium and 120 h after transfection from cells and the medium. Viral particles from the medium were precipitated with 40% polyethylene glycol 8000 (Acros Organics, Germany) in 500 mM NaCl for 2 h at 4 °C and then after centrifugation at $4000 \times g$ for 30 min combined with cell pellets for processing. The cell pellets were suspended in 500 mM NaCl, 40 mM Tris, 2.5 mM MgCl₂, pH 8, and 100 U ml⁻¹ of salt-activated nuclease (Arcticzymes, USA) at 37 °C for 30 min. Afterwards, the cell lysates were clarified by centrifugation at $2000 \times g$ for 10 min and then purified over iodixanol (Optiprep, Axis Shield, Norway) step gradients (15%, 25%, 40%, and 60%)[37,38] at $350,000 \times g$ for 2.25 h. Viruses were concentrated using Amicon filters (EMD, UFC910024) and formulated in sterile phosphate-buffered saline (PBS) supplemented with 0.001% Pluronic F-68 (Gibco, Germany). Virus titers were measured using AAV titration kit (TaKaRa/Clontech) according to manufacturer's instructions by determining the number of DNase I-resistant vg using qPCR (StepOne, Applied Biosystems). Purity of produced viruses was routinely checked by silver staining (Pierce, Germany) after gel electrophoresis (Novex™ 4–12% Tris–Glycine, Thermo Fisher Scientific) according to manufacturer's instruction. The presence of viral capsid proteins was positively confirmed in all virus preparations. Viral stocks were kept at −80 °C until injection.

**Virus injections**. Viral vectors used in this study were either AAV-2/6 or the recently engineered AAV-PHP.B[36]. Vectors carried plasmids that code for the Channelrhodopsin-2-variant CatCh linked to the reporter-protein eYFP under control of the human synapsin promotor (titer: AAV2/6: $3.2 \times 10^{12}$–$2.7 \times 10^{13}$ GC/ml; PHP.B: $4.57 \times 10^{12}$ GC/ml). Injections were performed using micropipettes (20 μm tip diameter) pulled from quartz capillaries on a P-2000 laser puller (Sutter Instruments) connected to a pressure microinjector (100–125 PSI, PLI-100 pico-injector, Harvard Apparatus). 2–3 μl of virus suspension were injected directly into the left spiral ganglion of adult gerbils under general anesthesia using a recently developed intramodiolar approach[20,39]. After making an incision behind the ear, muscles and connective tissue covering the bulla tympanica were displaced and a bullostomy was performed in order to access the cochlea. Using a KFlex dental file, a small hole was then drilled into the basal part of the modiolus via the dorsal part of the round window niche to directly access the spiral ganglion. After injection, muscles and connective tissue were repositioned and the skin was sutured. Animals were allowed to recover for at least 4 weeks after surgery before continuing experiments. Positively transfected animals showed no significant differences in both thresholds (AAV2/6: $3.49 \pm 3.03$ mW, PHP.B: $1.70 \pm 0.99$ mW, mean/SD, $p = 0.073$, two-sample $t$-test) and maximal strength of responses (AAV2/6: $4.27 \pm 0.80$ $d'$ values, PHP.B: $4.55 \pm 1.11$ $d'$ values, mean/SD, $p = 0.57$, two-sample $t$-test) that could be evoked in the IC dependent on the virus they were injected with (Supplementary Fig. 13).

**Stimulation**. Stimuli were generated and presented via a custom-made system based on NI-DAQ-Cards (NI PCI-6229; National Instruments; Austin, USA) controlled with custom-written MATLAB scripts (The MathWorks, Inc.; Natick, USA). Acoustic stimuli were presented near field via a loudspeaker (Scanspeak Ultrasound; Avisoft Bioacoustics, Glienicke, Germany) centered 30 cm in front of the animals' head. A 0.25-inch microphone and measurement amplifier (D4039; 2610; Brüel & Kjaer GmbH, Naerum, Denmark) were used to calibrate sound pressure levels. For optical stimulation, access to the cochlea was achieved using the surgical approach described for virus injections. An optical fiber (200 μm diameter, 0.39 NA; Thorlabs, Dachau, Germany) coupled to a blue laser (473 nm, 100 mW DPSS; Changchun New Industry Optoelectronics) was then inserted into the cochlea either via the round window (for basal stimulation) or via cochleostomies in the middle or apical cochlear turn, respectively (stimulation sites in which the cochleostomy resulted in bleeding have been excluded from analysis; 3/29 STCs). Radiant flux from the fiber aperture was calibrated with a power meter before each experiment (Solo-2; Gentec-EO; München, Germany). Biphasic pulses (100 μs phase duration) of varying electric current were generated with a custom-made current-source stimulus isolator and delivered via four-channel (600 μm electrode spacing) rodent eCI, provided by Roland Hessler, MED-EL Innsbruck (for details see ref. [40]). The implant was inserted into the scala tympani via the round window, such that the most apical electrode was ~5–6 mm within the cochlea. Implant positioning was confirmed by our physiological results: Considering 11 mm length of the gerbil basilar membrane and a hearing range of ~0.2–50 kHz, the eCI would cover ~5.5/11 mm (=50%) of the cochlear length, which corresponds—according to the Greenwood function—to a CF at the cochlear location at the tip of the eCI of approximately $f = 0.39(10^{2.1x} - 0.5) = 4.18$ kHz[28,41]. The mean CF recorded in response to electrical stimulation at electrode 1 (which is located at the very tip of the implant) was 3.72 kHz in response to monopolar and 4.5 kHz in response to bipolar stimulation (Supplementary Fig. 7). The return electrode for monopolar electrical stimulation was placed between connective tissue and the bone outside of the bulla tympanica. The return electrode for bipolar stimulation was chosen as the neighboring electrode to the stimulation electrode in basal direction.

**Recording of MUA**. To access the IC, an incision was made in the animal's scalp along the midline of the skull. After cleaning the bone, a thin layer of self-etching

UV glue (Orbi-Bond; Orbis Dental, Münster, Germany) was applied and a head post was mounted rostrally to bregma using dental cement (Paladur; Kulzer, Hanau, Germany). The animal's head was fixed and bregma and lambda were aligned stereotactically. A low impedance (<1 Ω) metal wire was implanted between the skull and the cortical surface on the left hemisphere to serve as a reference electrode. Using a dental drill, a craniotomy (~1 mm diameter; centered 2 mm lateral and 0.5 mm caudal to lambda) was performed on the right hemisphere of the animal's skull in order to access the IC contralateral to the injected ear. After removing the dura over primary visual cortex (which partly covers the gerbil's IC[42]) with a sharp needle, a linear 32-electrode silicon probe (177 μm² electrode surface, 50 μm electrode spacing, 1–3 MΩ impedance measured at 1 kHz; Neuronexus, Ann Arbor, USA) was positioned above the brain ~2 mm lateral to lambda and as close as possible to the transverse sinus (which also covers the IC in this species) as possible. Initially, the probe was slowly inserted ~3.3 mm into the brain (measured from the surface of visual cortex) using a LN Junior 4RE micromanipulator (Luigs & Neumann; Ratingen, Germany). After waiting for 1 h in order to obtain a stable preparation, a first mapping of recording sites was done using acoustic stimuli at 60–80 dB ranging from 0.5 to 32 kHz. Based on the measured neuronal activity the probe was then further advanced (or retracted) in order to optimally access the tonotopic axis of the central nucleus of the IC[24,43] and to be able to compare neuronal responses to optogenetic/electrical stimulation across animals later on. Once the silicon probe was positioned, activity of multi-neuronal clusters was amplified, filtered (0.1–9000 Hz) and recorded at a sampling rate of 32 kHz using a Digital Lynx 4S recording system (Neuralynx; Dublin, Ireland). Data was stored on a hard drive and analyzed off-line.

**Event extraction**. All data was analyzed using custom-written MATLAB scripts. To obtain time stamps of neuronal events, thresholds were set manually on 0.6–6 kHz bandpass-filtered (fourth-order Butterworth filter) data traces, usually at the level of three times the median absolute deviation of the whole data trace. Each crossing of this threshold was considered a time stamp of a neuronal event and a refractory time of 1 ms was implemented after each time stamp. For electrical stimulation, a linear interpolation was performed from the sampling point just before trigger onset to the sampling point 3 ms after trigger onset before thresholding in order to remove the electrical artifacts from the data trace (Supplementary Fig. 14a–c). The artifact removal has subsequently been verified with data recorded from dead animals where no neuronal component could be observed in addition to the artifacts (Supplementary Fig. 14d–h).

**Frequency tuning**. Frequency response areas were constructed in response to pure tones (100 ms duration, 5 ms sine squared ramps for stimulus onset and offset, 150 ms inter-stimulus interval) ranging from 0.5 to 32 kHz in quarter octave steps at sound pressure levels ranging from 10 to 80 dB SPL in 10 dB steps. 20–30 repetitions of each frequency–intensity combination were presented in a pseudo-random order, where each stimulus was presented once before presenting the next iteration of trials. The characteristic frequency (CF) and its corresponding threshold at each recording site was determined as the frequency that elicited responses at the lowest sound pressure level during the period of stimulus presentation[44,45].

**Responses to artificial SGN stimulation**. To determine a time window in which potential responses to optical or electrical stimulation can be evaluated, a PSTH (0.05 ms bin size) was constructed from pooled MUA recorded at all recording sites in response to a 1 ms laser pulse or a biphasic pulse of electrical current (100 μs phase duration) with the maximal stimulus intensity presented (~35 mW, 500 μA). The response was defined as MUA exceeding the mean spike rate plus three standard deviations 20 ms before stimulus onset and was observed 2.25–24.5 ms after stimulus onset for optical stimulation and 2.45–13.25 ms after stimulus onset for electrical stimulation (Supplementary Fig. 2). Based on these results, the response windows were set to 0–25 ms after stimulus onset for optical and to 0–14 ms after stimulus onset for electrical stimulation.

**Spatial spread of excitation**. To quantify response thresholds and the spread of excitation, spatial tuning curves (STC) based on the cumulative discrimination index ($d'$) of spike rates were constructed[7,21,22]: For each electrode, the distribution of spikes during 20–30 trials in response to one stimulus was compared against the distribution of spikes during each trial in response to the subsequent (higher intensity) stimulus, where stimuli are sorted according to their intensity, starting with an intensity of zero (no stimulus condition). A receiver operating characteristic (ROC) curve was constructed from these distributions. The area under the ROC curve, which depicts the $Z$-score (measured in standard deviations), was then multiplied by $\sqrt{2}$ in order to obtain the $d'$ value. $d'$ values of successively increasing stimulus intensities were finally summed up in order to calculate the cumulative discrimination index. In the next step, a matrix was constructed by sorting cumulative $d'$ values according to the electrode position they were obtained from in one dimension and the stimulus intensity they were evoked by in the other dimension. Iso-$d'$-contour-lines were then interpolated by using MATLABs built-in contour function and thresholds of neuronal activation were determined as the stimulus intensities that correspond to isolines at the cumulative $d'$ level of 1. The best electrode (BE) was defined as the electrode which showed the lowest threshold

($d' = 1$). The spread of excitation was accessed as the distance between the most dorsal and the most ventral electrode with a $d' \geq 1$ at the stimulus intensity that elicited a $d'$ value of 1.5, 2, 2.5, or 3 at the BE.

Monte Carlo ray tracing: Monte Carlo ray tracing simulation was performed using TracePro® Standard 7.8.1 (Lambda Research Corporation) to validate the experimentally used fiber stimulation. Briefly, different cochlear compartments were reconstructed from x-ray tomography and embedded in a solid cube to simulate bone enclosure. Each of the structures was assigned with mean optical properties from cerebrospinal fluid, brain tissue, and bone, respectively[20]. All the light sources were modeled as the optical fiber used in the experiments (Thorlabs FT200UMT, 0.39 NA) and defined in TracePro as grid sources with the following parameters: circular pattern of 3003001 rays (1001 rings, $\lambda = 473$ nm, uniform total intensity of 10 mW); grid boundary radius: 100 μm; symmetric Gaussian spatial and angular beam distribution (waist radius of Gaussian beam profile: 100 μm; half angle of angular profile of the beam: 16.79°). Emitter surfaces were calculated for every position: First, the tonotopic axis was defined along the center line of Rosenthal's canal, where 300 query points were extrapolated and the corresponding frequency positions were calculated by the Greenwood function for a hearing range of 50–0.2 kHz ($f = 0.39(10^{2.1x} - 0.5)$, where $x$ is the cochlear length normalized from 0 to 1 in baso-apical direction). Second, coordinates for the tips of optical fibers were placed in anatomically meaningful positions corresponding to fiber placement in our in vivo experiments. These coordinates were then translated following a straight line to Rosenthal's canal to a given distance from the query point (400, 700, and 900 μm for apical, mid-cochlear, and basal stimulation, respectively). The newly calculated coordinate was defined as the origin of the light source and the straight line as its normal vector. To account for variability in fiber placement during our experiments, two models were calculated. In the rotation model, the normal vector was rotated ±15° in two perpendicular planes. In the translational model, the normal vector was coaxially translated ±100 and 200 μm. The origin and the normal vector of all five sources (initial position plus four rotations or four translations) at the three positions were imported to TracePro. Radiant flux was read from the 300 query points (included as solid spheres with a 5 μm radius, with assigned optical properties of brain tissue). Irradiance was calculated as radiant flux/4*pi*radius². Irradiance values were linearly scaled to 2.67 mW, which was the mean threshold for optogenetic stimulation observed in our experiments (Supplementary Fig. 5). The mean irradiance profile was calculated for every position using the irradiance profile of all five sources, and the tonotopic location the fiber was facing was then calculated as the peak of the mean irradiance profile.

**IC histology**. After each experiment the silicon probe was retracted with the micromanipulator and a tungsten electrode covered with DiI (DiIC18(3); Thermo Fisher Scientific, Darmstadt, Germany) was inserted into the IC at the same site in order to stain the electrodes' position. Afterwards brains were explanted and fixed in 4% PFA in phosphate buffered saline (PBS) for several days before they were moved to 30% sucrose in PBS for cryoprotection. Coronal slices of 50 μm thickness were obtained using a Leica CM3050 Cryostate (−25 °C object temperature). Slices were mounted on microscope slides using Fluoroshield mounting medium (Sigma Aldrich, Darmstadt, Germany), which contains DAPI in order to stain the cell nuclei.

**Reporting summary**. Further information on research design is available in the Nature Research Reporting Summary linked to this article.

## Data availability
Data generated and analyzed during the current study is available from the corresponding author upon reasonable request. Numerical source data underlying Figs. 1e, 2a–k, 4b–j, as well as supplementary figs. 1, 2, 5, 7, 9, 11, 12, 13 are provided as a source data file.

## Code availability
Analysis code—written in MATLAB 2016a—is available from the corresponding author upon reasonable request.

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

## Acknowledgements

The authors are very grateful to Carolyn Garnham and Roland Hessler from MED-EL for providing cochlear implants for animal research. We thank Gerhard Hoch for hardware and software development of multi-channel electrical stimulation and expert technical assistance. We thank Daniel Keppeler for providing the x-ray images in supplementary fig. S2 and providing an x-ray tomography reconstructed gerbil cochlea for Monte-Carlo ray tracing. We thank Daniela Gerke for expert help with virus preparation. We thank Ulrich Schwarz for discussion regarding the dynamic range estimations. We thank Ben Deverman and Viviana Gradinaru for providing the PHP.B construct used in this study. The work was funded by the European Research Council (ERC) under the European Union's Horizon 2020 research and innovation program (grant agreement no. 670759—advanced grant "OptoHear") to T.M. A.D. is a fellow of the German Academic Scholarship Foundation.

## Author contributions

A.D., M.J., and T.M. designed the study. M.J. designed hardware and software for multi-channel electrophysiological recordings. A.D. performed virus injections, IC recordings and IC histology. C.J.D.-A. performed Monte-Carlo ray trace modeling. V.R. produced the PHP.B-CatCh virus. A.D. performed data analysis under the supervision of M.J. and T.M. A.D., M.J., and T.M. prepared the manuscript.

## Additional information

**Competing interests:** The authors declare no competing interests.

