## [Peer Review File · Nature Communications]

Reviewers' Comments:

Reviewer #1:

Remarks to the Author:

This is a highly significant study demonstrating that optical stimulation of the cochlea can provide spectral selectivity approaching that of normal acoustical stimulation and far surpassing that obtainable with monopolar electrical cochlear implant stimulation. This follows on previous innovative work from the Moser laboratory that demonstrated the efficacy of optical stimulation, but this is the first to demonstrate spectral selectivity in such a compelling and quantitative manner. I have only a few suggestions intended to improve the clarity of presentation in this largely well written manuscript.

Figures 1E and S1: The ordinate is given as depth from the cortical surface. Please clarify that this is (I think) relative to the surface of overlying occipital cortex, or maybe the cortex of the IC. Or, preferably, give this in terms of recording electrode number to facilitate comparison with Figures 3, 4A-D, S3, and S4.

Lines 237-239: The placement of the reference to Figs. 4D and 5B after "even though it has been observed in some animals" seems to indicate that those figures show tonotopic activation, whereas I think that the figures show the absence of tonotopic activation, as stated in the beginning of the sentence.

Lines 269-271, "Few units in ventral ICC...": This seems to indicate that you were expecting to find low frequency units in ventral ICC whereas only a few were found. I think that what is meant is that "A few units in ventral ICC...", indicating that this was an unexpected finding.

Lines 328-329: Again, the order of the clauses is confusing. This seems to indicate that it is the intracochlear oCI that is not directly feasible for clinical translation, whereas I think that what is meant is that the multiple laser-coupled fibers are not clinically feasible.

Line 475: I think that "subsequent stimulus" would be clearer as "subsequent (higher intensity) stimulus". As it is, "subsequent" only makes sense after the sorting, not necessarily subsequent in time.

Figure S1: Something is missing from the end of the legend.

Figure S2: "Blue lines" should be "Blue and orange lines". Please clarify whether these histograms are from single units or compiled across some neural population (and give the N).

John C. Middlebrooks
University of California at Irvine

Reviewer #2:

Remarks to the Author:

In the present manuscript Dieter et al. compare spectral (sound) selectivity of natural, electrical and optogenetic cochlear stimulation in gerbils. This is done by measuring tonotopy in the inferior colliculus with linear extracellular electrodes. Since the cochlea is also tonotopically organized, the location where inner hair cells and downstream spiral ganglion neurons are activated determines the perceived sound frequency. If hearing is lost, for example due to hair cell damage or loss, stimulation of spiral ganglion neurons can restore hearing. In theory, precise tonotopic stimulation with high spatial resolution should enable restoration of the full frequency spectrum with high spectral selectivity. However, electrical cochlear implants are limited with respect to the latter. Electrical stimulation is spreading laterally, thereby activating spiral ganglion neurons outside the

desired frequency spectrum. This cross-talk of electrodes limits the precise perception of acoustic signals from the environment.

Here the authors argue that optogenetic activation of spiral ganglion neurons could remedy this limitation of existing cochlear implants. They optically stimulate spiral ganglion neurons virally transduced with the rhodopsin CatCh at three different positions in the cochlea. This is achieved by inserting optical fibers in the apical and mid turn and in the round window of the cochlea. To measure spectral selectivity, they record light-evoked responses in the inferior colliculus.

The main (and only) new finding of this manuscript is that optogenetic stimulation of spiral ganglion neurons appears to be tonotopically represented in the inferior colliculus with higher accuracy than stimulation via monopolar electrical cochlear implant. Since (sound) frequency specificity is not well preserved with monopolar electrical cochlear implants, they conclude that optical cochlear implants combined with expression of excitatory rhodopsins in spiral ganglion cells, could potentially overcome this problem. While the dataset presented in this manuscript is convincing, I see some major conceptual problems with this study, which tends to overemphasize the advantages of the optogenetic approach, neglecting alternative methods of electrical stimulation and potential flaws and drawbacks associated with optogenetic stimulation. My main concern relates to the localization of electrical vs. optical stimulation sites and to the way electrical stimulation is done.

1. No quantification is presented, where either electrical cochlea implants or optical fibers were placed in individual experiments. Due to the tonotopic organization of the cochlea, it is important to verify where exactly the stimulation happened and if the localization of electrodes and optic fibers was similar between animals. For example, it is difficult to explain why in case of optogenetic stimulation in some animals the characteristic frequency decreased in the apical-basal direction (Fig. S5). It should increase. Was this due to misplacement of the fibers or due to a large spectral spread or something else? In addition some electrodes show much better reproducibility of characteristic frequency between animals compared to optogenetic stimulation (Fig. S5). Especially, optogenetic mid-turn and base stimulation showed variability of > 10 octaves in the characteristic frequency, indicating that despite lower spectral spread, reproducibility of optogenetic CI stimulation is limited by a strong variability of the characteristic frequency.

2. Related to point 1, n values are low for the apical stimulation site in panel A of figure S5 and they do not match the numbers of experiments presented in Fig. 4C (which is a different analysis of the same dataset). Why were some experiments omitted in fig. S5? Similarly, two data points are missing in Fig. 4C (total STCs = 26) compared to 4F (n = 28 STCs). Since 4F is based on the same dataset as 4C, it is not clear to me why two more STCs appear in 4F. In contrast, n numbers are consistent for electrical stimulation across all panels. Why were data from the optogenetic experiments omitted in some analyses and in others not? The authors should show the missing values.

3. The distance between the stimulation sites in the eCI and oCI preparations appear to differ quite substantially. The distance between each of the four electrode channels used here to evaluate spectral spread was 600 μm . Thus, the entire distance covered, was 1.8 mm, which is approx. 16 % of the entire cochlear length (11 mm, see e.g. Dong et al. Biophys J 2009). This small spacing also seems to be represented in Figure 3C, where, despite the broad tuning curves, the range of the best electrodes spreads only over 10 recording electrodes in IC between stimulation electrodes 1 and 4. In contrast, optical fibers were inserted at three positions extending over a major portion of the cochlear length (round window, mid-turn, apical turn). Thus, distance between two fibers is most likely several mm. However, exact numbers are missing since the authors do not provide enough detail. Thus, despite larger spectral spread of electrical stimulation, optogenetic stimulation was biased towards more spatially separated stimulation sites and therefore it is not clear to what extent the lacking tonotopy of electrical stimulation can be simply attributed to the narrow spacing of the stimulation electrodes.

4. Most important, the authors only used monopolar electrical stimulation, which results in much less precise tonotopic responses compared to bipolar stimulation (e.g. van den Honert 1987, Zhu 2012). Technically, bipolar stimulation is relatively easy to achieve with the electrode arrays used here. Although the authors discuss the advantage of bipolar (and also nerve-penetrating) electrical stimulation, they do not acknowledge that bipolar stimulation will come very close to the optogenetic approach. They calculate that optogenetic stimulation increases spread 1.61-fold and monopolar stimulation increases spread 4.03-fold with respect to acoustic stimulation. Considering that bipolar stimulation is spreading approx. half as wide as monopolar stimulation (0.5-fold if considering the numbers from Middlebrooks, 2007, which are discussed in this manuscript), bipolar electrical stimulation would be expected to spread only 2.04-fold compared to acoustic stimulation. This spread would be roughly similar to the 1.61-fold spread with optogenetics. Since the authors make a big point about the advantage of optogenetic stimulation for future prosthetic devices, they need to compare the spread of excitation to the best and not the worst electrical method with respect to spreading of excitation. Thus, at least bipolar stimulation has to be measured in the same configuration as monopolar & optogenetic stimulation.

Minor points:

1. "neuroprothesis" should be "neuroprosthesis"
2. Fig. 1D: rotate panels 90 degrees CW to match orientation of panels in fig. 3
3. Fig. 3: labeling of y axes appears incorrect. electrode "2" should be "4".
4. In the main text the authors refer to figure S10 which is missing in the supplement.
5. Figure S1: In the legend the authors refer to "lowest" and "highest" slopes. Slopes are steep or shallow, but not high or low.
6. Figure S4: Text is missing in the figure legend.
7. Figures S5 & S7: Other than indicated in the legend, the mean values are not plotted.
8. Discussion. The authors argue that red-shifted opsins will allow for more precise stimulation due to "less tissue-scattering of longer wavelength light". The opposite is true. Scattering and, more importantly, absorption of short-wavelength light lead to a steeper fall-off of light intensity with increasing distance from the light source. Thus, blue light is more restricted in space and does not penetrate as deeply as red light. Hence, activation of opsins is more spatially confined with blue-shifted light (see e.g. Stujenske 2015).

Responses to reviewer comments of the manuscript

“Near physiological spectral selectivity of cochlear optogenetics”

Alexander Dieter, Dr. Vladan Rankovic, Dr. Marcus Jeschke and Prof. Tobias Moser

Reviewer #1 (Remarks to the Author):

This is a highly significant study demonstrating that optical stimulation of the cochlea can provide spectral selectivity approaching that of normal acoustical stimulation and far surpassing that obtainable with monopolar electrical cochlear implant stimulation. This follows on previous innovative work from the Moser laboratory that demonstrated the efficacy of optical stimulation, but this is the first to demonstrate spectral selectivity in such a compelling and quantitative manner. I have only a few suggestions intended to improve the clarity of presentation in this largely well written manuscript.

We are very grateful to Prof. Middlebrooks for the appreciation of our work, as well as for his constructive comments that help us to further improve and clarify our manuscript.

Figures 1E and S1: The ordinate is given as depth from the cortical surface. Please clarify that this is (I think) relative to the surface of overlying occipital cortex, or maybe the cortex of the IC. Or, preferably, give this in terms of recording electrode number to facilitate comparison with Figures 3, 4A-D, S3, and S4.

Done. In response to the comment, we clarified that the depth indicated on the ordinate is relative to occipital cortex for Figs. 1 and S1. We decided to refrain from changing the ordinate to indicate recording electrode number because the experimental variation of insertion depth among the recorded animals leads to slight differences in electrode positions across our data.

Lines 237-239: The placement of the reference to Figs. 4D and 5B after “even though it has been observed in some animals” seems to indicate that those figures show tonotopic activation, whereas I think that the figures show the absence of tonotopic activation, as stated in the beginning of the sentence.

It is correct that the calculated correlation did not show tonotopic activation. The statement “even though it has been observed in some animals” refers to the fact that in some animals (shown by individual symbols) the best electrode showed a dependence on the stimulation electrode. These were not significant on the population level, but we decided to point out individual examples in order to not undersell the capabilities of electrical CIs. However, in the revised manuscript - including twice as much data for monopolar electrical stimulation - we observe a significant correlation on the population level and thus rephrased our statement.

Lines 269-271, “Few units in ventral ICC...”: This seems to indicate that you were expecting to find low frequency units in ventral ICC whereas only a few were found. I think that what is meant is that “A few units in ventral ICC...”, indicating that this was an unexpected finding.

Correct. We have rephrased to improve clarity.

Lines 328-329: Again, the order of the clauses is confusing. This seems to indicate that it is the intracochlear oCI that is not directly feasible for clinical translation, whereas I think that what is meant is that the multiple laser-coupled fibers are not clinically feasible.

Again, this is what we meant, thanks for pointing it out. We have rephrased this sentence.

Line 475: I think that “subsequent stimulus” would be clearer as “subsequent (higher intensity) stimulus”. As it is, “subsequent” only makes sense after the sorting, not necessarily subsequent in time.

Good point. We have added the “(higher intensity)” in order to avoid confusion.

Figure S1: Something is missing from the end of the legend.

You are right. Also in other supplemental figures this was the case (probably this happened at some point when formatting our supplementary material). We have now added the supplemental figure legends in the correct way.

Figure S2: “Blue lines” should be “Blue and orange lines”. Please clarify whether these histograms are from single units or compiled across some neural population (and give the N).

“Orange lines” was added to complete the legend. Also, we have stated that we compiled the PSTHs across all multi-units that have been recorded in response to optical or electrical stimulation, respectively. The N is now stated.

John C. Middlebrooks
University of California at Irvine

Reviewer #2 (Remarks to the Author):

In the present manuscript Dieter et al. compare spectral (sound) selectivity of natural, electrical and optogenetic cochlear stimulation in gerbils. This is done by measuring tonotopy in the inferior colliculus with linear extracellular electrodes. Since the cochlea is also tonotopically organized, the location where inner hair cells and downstream spiral ganglion neurons are activated determines the perceived sound frequency. If hearing is lost, for example due to hair cell damage or loss, stimulation of spiral ganglion neurons can restore hearing. In theory, precise tonotopic stimulation with high spatial resolution should enable restoration of the full frequency spectrum with high spectral selectivity. However, electrical cochlear implants are limited with respect to the latter. Electrical stimulation is spreading laterally, thereby activating spiral ganglion neurons outside the desired frequency spectrum. This cross-talk of electrodes limits the precise perception of acoustic signals from the environment.

Here the authors argue that optogenetic activation of spiral ganglion neurons could remedy this limitation of existing cochlear implants. They optically stimulate spiral ganglion neurons virally transduced with the rhodopsin CatCh at three different positions in the cochlea. This is achieved by inserting optical fibers in the apical and mid turn and in the round window of the cochlea. To measure spectral selectivity, they record light-evoked responses in the inferior colliculus.

The main (and only) new finding of this manuscript is that optogenetic stimulation of spiral ganglion neurons appears to be tonotopically represented in the inferior colliculus with higher accuracy than stimulation via monopolar electrical cochlear implant. Since (sound) frequency specificity is not well preserved with monopolar electrical cochlear implants, they conclude that optical cochlear implants combined with expression of excitatory rhodopsins in spiral ganglion cells, could potentially

overcome this problem. While the dataset presented in this manuscript is convincing, I see some major conceptual problems with this study, which tends to overemphasize the advantages of the optogenetic approach, neglecting alternative methods of electrical stimulation and potential flaws and drawbacks associated with optogenetic stimulation.

My main concern relates to the localization of electrical vs. optical stimulation sites and to the way electrical stimulation is done.

First, we would like to thank the reviewer for acknowledging that our dataset presented in the manuscript is convincing. The constructive evaluation of our work has helped us to further improve the manuscript. We have addressed all concerns raised by the reviewer and improved the manuscript both by including new data and with rephrasing the MS in order to avoid any apparent overemphasizing of advantages of optogenetic over electric cochlear stimulation. Most notably, as requested, we have performed additional experiments employing bipolar electrical stimulation.

1. No quantification is presented, where either electrical cochlea implants or optical fibers were placed in individual experiments. Due to the tonotopic organization of the cochlea, it is important to verify where exactly the stimulation happened and if the localization of electrodes and optic fibers was similar between animals.

We agree with the reviewer that careful placement of optical fibers and electrical CIs is critical for meaningful data interpretation. While the exact quantification of implant placement in each animal was not possible for technical reasons, we have aimed to avoid overt variation of implant positions across different animals and to estimate the positions as follows:

(1) In the case of electrical stimulation, we measured the insertion depth of the implant into the scala tympani of each animal. Little silicone blobs on the implant, made to enable easier implantation, helped us to determine the insertion depth, which was between 5-6 mm (as stated in the methods section). The stated positioning of our eCIs is consistent with the literature: Considering 11 mm length of the gerbil basilar membrane (as indicated by reviewer 2; Dong et al. Biophys J 2009) and the hearing range of gerbils (~ 0.2 -50 kHz; i.e. 7.97 octaves; e.g. Huet et al. Hear Res 2016), we calculate a tonotopic slope of 1380 $\mu\text{m}/\text{octave}$ in the cochlea (consistent with the slope of 1500 $\mu\text{m}/\text{octave}$ in the higher frequency range (< 4 kHz) reported by Müller Hear Res 1996). With an average eCI insertion depth of ~ 5 mm we would cover 45.45% of the cochlear length, corresponding to 3.62 octaves. The characteristic frequency at this location should therefore be $\sim 2^{(-3.62)} * 50\text{kHz} = \mathbf{4.07\text{ kHz}}$. The mean characteristic frequency at the best electrode obtained with electrical stimulation at electrode 1 (which is located at the very tip of the implant) was 3.72 kHz in response to monopolar and 4.5 kHz in response to bipolar stimulation (fig. S5). These values fall pretty well into the expected range and thereby confirming the positioning of our electrical CI. We have also now expanded the statement on eCI positioning in the method section.

(2) In the case of optical stimulation, estimation of fiber positioning is not as straightforward as in the case of electrical stimulation. In order to respond to the reviewer's comment and to increase confidence into the positioning of the optical fibers, we have now performed Monte Carlo simulations modeling of optical rays (similar to the one we have already published in Wrobel & Dieter et al. Sci Trans Med 2018). We employed a reconstruction of a gerbil cochlea (based on x-ray phase-contrast tomography), where the place-frequency-map of the spiral ganglion has been fitted with the Greenwood-function, considering the gerbil hearing range of 0.2-50 kHz. Properties of the optical fiber have been adapted from the datasheets of fibers employed in our experiments and optical properties of different tissues were taken from literature. We then positioned optical fibers in

an anatomically meaningful way as during our physiology experiments (via the round window, middle and apical cochlear turn). 3D Monte Carlo ray tracing (TracePro software) was performed modeling ~3 million optical rays. We then took the SGN regions with the highest illumination values as the primary regions of neural excitation. The values obtained from this model match our physiology results quite well, hence, confirming the fiber positions estimated from IC recordings. These results were added as fig. S8.

For example, it is difficult to explain why in case of optogenetic stimulation in some animals the characteristic frequency decreased in the apical-basal direction (Fig. S5). It should increase. Was this due to misplacement of the fibers or due to a large spectral spread or something else?

We agree with the reviewer that the characteristic frequency (CF) should increase in the apical-basal direction within the ICC and, in fact, it does so in most of the cases. There are, however, two animals in which the CF decreases when comparing mid-cochlear stimulation to stimulation via the round window. This was neither due to misplacement of the fibers nor to larger spectral spread than usual. This view is supported by comparison of the best electrodes (BE) in response to stimulation at the same locations in Figure 4 C: BE strictly increases in apical-basal direction. Instead, in these two animals the best electrodes in the ventral inferior colliculus responded to low-frequency stimulation stronger than it did to high frequency stimulation, most likely because these electrodes were positioned outside the ICC or in close proximity to the ICC border. However, the focus of excitation still was the ventral ICC region, where one would expect it to be. We have encountered such apparently deviating responses with the most-ventrally positioned electrodes also with acoustic stimulation (see Figure 1) and this observation is consistent with literature reports (Schnupp et al., Front Neural Circuits 2015). In response to the reviewers comment we have now enhanced the discussion of this finding (beginning of the discussion section) and also added an explanatory statement to the legend of fig. S7.

In addition some electrodes show much better reproducibility of characteristic frequency between animals compared to optogenetic stimulation (Fig. S5). Especially, optogenetic mid-turn and base stimulation showed variability of > 10 octaves in the characteristic frequency, indicating that despite lower spectral spread, reproducibility of optogenetic CI stimulation is limited by a strong variability of the characteristic frequency.

We can see how the reviewer arrived at this impression, but we respectfully disagree that this concern is substantiated by the data: Mean \pm SD of CFs are 0.63 ± 0.15 kHz for apical, 3.5 ± 3.23 kHz for mid-cochlear and 13.1 ± 9.5 kHz for basal stimulation. The coefficient of variation (SD/mean) of the CF, the standard deviations are 0.24, 0.92 and 0.7, respectively. It is true that amongst these, the apical stimulation is the least variable. However, also the coefficient of variation for mid-cochlear and round window stimulation is below 1. The display of the standard deviations on the logarithmic scale “seems” to extend the error bars towards the lower end of the scale. Furthermore, the reproducibility of optogenetic SGN stimulation can be seen in Fig. 4C, where we show the BE in dependence of stimulation site.

2. Related to point 1, n values are low for the apical stimulation site in panel A of figure S5 and they do not match the numbers of experiments presented in Fig. 4C (which is a different analysis of the same dataset). Why were some experiments omitted in fig. S5?

We appreciate that point and we should have explained it in more detail. The underlying reason for the discrepancy in the number of data points is that, when performing cochlear surgery in order to place optical fibers or electrical CIs, the auditory thresholds of multi-units in the contralateral ICC increase by ~ 20 dB (30.3 dB SPL vs 48.1 dB SPL, before and after surgery, respectively; now shown in new fig. S2). Considering the audiogram of the gerbil, neurons responding to frequencies < 2 kHz and > 8 kHz have higher thresholds than units responding to ~ 2 -8 kHz (fig. S2). As stated before, by performing cochlear surgery, the whole audiogram shifts by ~ 20 dB (also shown in fig. S2). We could not determine the characteristic frequency (CF) for all the electrodes we recorded from, probably because after surgery some of the units had thresholds above 80 dB SPL, beyond the limit of our speaker's output in the range of 0.5-32 kHz. Thus, we could still determine a best electrode (BE) of these units (as shown in Fig. 4C) but could not determine the CF of this electrode (fig. S7), explaining the discrepancy in numbers of data points. Since the BEs in response to apical and basal optical stimulation are at the dorsal and ventral edges of the ICC, where low and high frequencies, respectively, are coded, we could not determine the CFs in the case of three animals for apical stimulation and one case of basal stimulation. For mid-cochlear stimulation, as well as for electrical stimulation (where electrodes were placed – as discussed before – quite in the center of the cochlea (5/11 mm)), this was not the case. This is readily explained by the lower thresholds for the mid-frequencies coded such that we could still evoke auditory activity in these regions after performing cochlear surgery. In order to make this more evident to the reader we have now included fig. S2 and added an explanation to the legend of figure S7. Furthermore, in response to the reviewer's comment we now analyzed the spread of excitation not only as spectral spread, i.e. when spatial spread is normalized to the tonotopic axis in each animal, but also as spatial spread, i.e. activation of a given anatomical region in the ICC, which we now present in supplemental figure S9.

Similarly, two data points are missing in Fig. 4C (total STCs = 26) compared to 4F (n = 28 STCs). Since 4F is based on the same dataset as 4C, it is not clear to me why two more STCs appear in 4F. In contrast, n numbers are consistent for electrical stimulation across all panels. Why were data from the optogenetic experiments omitted in some analyses and in others not? The authors should show the missing values.

Once again, thanks for making this point. In this case, we realized that panel F in figure 4 had not been updated before initial submission and we sincerely apologize for this mistake. Out of all spatial tuning curves recorded for this dataset (n = 29) we had to exclude 3, due to damage upon cochlear surgery. We had stated in the methods "stimulation sites in which the cochleostomy resulted in bleeding have been excluded from analysis". In the figure 1 that we provide in this letter, we have plotted the threshold (in mW) and the maximum response strength (in d') for each of the 29 spatial tuning curves. It is evident from the figure, that 3 of these STCs had extraordinarily high thresholds and low response strengths (these ones are encircled in red). As we hope the reviewer will concur, these were the animals that we would consider excluding due to bleeding at the surgical site compromising the optophysiological response. Accordingly, panel 4C (n = 26) had already been updated in the initial submission, but unfortunately, we must have used a figure version where panel 4F was not updated. In 3F, 28 STCs are shown. The one missing to make it 29

Figure 1: Thresholds and maximum responses of STCs at the best electrode. Excluded animals are encircled in red.

was the one which you can see did not even reach a d' value of 1.5 in the STC, so there was no data point for spectral spread that could have been measured. To be precise, we now added that we had to exclude 3/29 STCs for the given the reasons in the method section. We hope that this sufficiently addresses the reviewer's concern and that the reviewer can agree on presenting only data of animals without surgical complications compromising the physiology.

3. The distance between the stimulation sites in the eCI and oCI preparations appear to differ quite substantially. The distance between each of the four electrode channels used here to evaluate spectral spread was 600 μm . Thus, the entire distance covered, was 1.8 mm, which is approx. 16 % of the entire cochlear length (11 mm, see e.g. Dong et al. Biophys J 2009). This small spacing also seems to be represented in Figure 3C, where, despite the broad tuning curves, the range of the best electrodes spreads only over 10 recording electrodes in IC between stimulation electrodes 1 and 4. In contrast, optical fibers were inserted at three positions extending over a major portion of the cochlear length (round window, mid-turn, apical turn). Thus, distance between two fibers is most likely several mm. However, exact numbers are missing since the authors do not provide enough detail. Thus, despite larger spectral spread of electrical stimulation, optogenetic stimulation was biased towards more spatially separated stimulation sites and therefore it is not clear to what extent the lacking tonotopy of electrical stimulation can be simply attributed to the narrow spacing of the stimulation electrodes.

We fully agree with the reviewer's notion on the importance of electrode positioning. Indeed, we would have preferred the electrode spacing of eCI to better accommodate the coverage obtained with optical fibers. We had also mentioned this point when presenting the results ("Note that the layout of the eCI might have contributed as 4 electrodes covered 1.8 mm and thus were not spread along the whole cochlea") and have now further emphasized the point. Moreover, as detailed in our response to the first comment we have enhanced the presentation regarding the placement of fibers and electrodes. However, better tonotopic activation of optogenetic vs electric stimulation was never mentioned as an advantage of optogenetic stimulation. In order to make this more evident to the reader, we have now separated the paragraph "Tonotopic activation of SGNs and spread of cochlear excitation with optogenetic stimulation" of the discussion into "tonotopic activation of SGNs" and "spread of excitation upon artificial SGN stimulation". In the first of these paragraphs we have now discussed the limitations of the electrode positioning and the implications for tonotopic activation of SGNs. Most importantly, however, with increasing the number of experiments we have now found a significant tonotopic activation for monopolar electrical stimulation despite the rather narrowly spaced electrodes.

4. Most important, the authors only used monopolar electrical stimulation, which results in much less precise tonotopic responses compared to bipolar stimulation (e.g. van den Honert 1987, Zhu 2012). Technically, bipolar stimulation is relatively easy to achieve with the electrode arrays used here. Although the authors discuss the advantage of bipolar (and also nerve-penetrating) electrical stimulation, they do not acknowledge that bipolar stimulation will come very close to the optogenetic approach. They calculate that optogenetic stimulation increases spread 1.61-fold and monopolar stimulation increases spread 4.03-fold with respect to acoustic stimulation. Considering that bipolar stimulation is spreading approx. half as wide as monopolar stimulation (0.5-fold if considering the numbers from Middlebrooks, 2007, which are discussed in this manuscript), bipolar electrical stimulation would be expected to spread only 2.04-fold compared to acoustic stimulation. This spread would be roughly similar to the 1.61-fold spread with optogenetics. Since the authors

make a big point about the advantage of optogenetic stimulation for future prosthetic devices, they need to compare the spread of excitation to the best and not the worst electrical method with respect to spreading of excitation. Thus, at least bipolar stimulation has to be measured in the same configuration as monopolar & optogenetic stimulation.

As was clear from the first submission, we recognize the importance of the work of the Middlebrooks and Snyder 2007 study, and were very pleased to see that one of the authors of this landmark paper, revealed his identity as reviewer (1) and endorsed our MS. Nonetheless, it is important to stress that monopolar stimulation is *the* most widely used modality of SGN stimulation in clinically used CI. Therefore, the comparison of optical stimulation to monopolar electrical stimulation is highly relevant. Nonetheless, in response to the request of reviewer 2 we have now performed further experiments and included data of six more gerbils in which we have measured IC responses to both mono- and bipolar electrical stimulation. At very low activation strengths (i.e. $d' = 1.5$) bipolar stimulation indeed outperformed monopolar stimulation and was not significantly different from acoustic and optogenetic stimulation. At higher activity levels there was no significant advantage of bipolar electrical stimulation over monopolar stimulation, and the spread of excitation was greater than for optogenetic and acoustic stimulation.

Hence, for the gerbil, we could not find the two-fold increase of spectral selectivity in response to bipolar stimulation over monopolar stimulation for higher stimulation levels previously reported for the cat by Middlebrooks and Snyder, 2007. We have addressed the discrepancy with literature values (Middlebrooks) primarily to the model system used: While the cat cochlea (used in that study) has a length of ~24 mm (Nadol, Hear Res, 1998) and a basal width of ~3 mm (Hatushika et al. Ann Otol Rhinol Laryngol, 1990), the gerbil cochlea has a length of ~11 mm (Dong et al. Biophys J 2009) and a width of ~1 mm (measured from X-ray tomography in our own lab). Thus, since the gerbil cochlea is smaller in size, a physically similar electric field originating from each eCI electrode might activate a larger fraction of the gerbil cochlea. If the bipolar field already excites almost the whole cochlea in the gerbil, there is little room for even more activation by using monopolar stimulation, whereas this difference would be much more pronounced in species with larger cochlea, e.g. cats or even humans. This interpretation is supported by the values we already calculated in the initial submission of our manuscript: compared to acoustic stimulation, Middlebrooks and Snyder, 2007 published a 4.16-fold increased spread of excitation upon bipolar and 8.23-fold spread of excitation upon monopolar electrical stimulation. In our case, the monopolar stimulation – which already excited the whole cochlea – was only 4.03-fold as large as the spread of excitation upon acoustic stimulation. Thus, we might have already underestimated monopolar stimulation (actually 2-fold when comparing to Middlebrooks and Snyder, 2007). We have not been able to find literature that has demonstrated improved spectral selectivity of bipolar vs monopolar stimulation in rodents. In conclusion, even though we could not demonstrate the advantage of bipolar over monopolar electrical stimulation at higher activation strengths in the gerbil, our data is highly valid to demonstrate the improvement in spectral resolution of optogenetic over (even bipolar) electrical stimulation of SGNs. We have added this data throughout the manuscript and carefully discussed the results. We are confident that we have pleased the reviewer's request with the data from these experiments and thereby could resolve an important concern on our present study.

Minor points:

1. "neuroprosthesis" should be "neuroprosthesis"

Corrected.

2. Fig. 1D: rotate panels 90 degrees CW to match orientation of panels in fig. 3

We would like to refrain from rotating these panels, since here we show frequency-response-areas (response strength as a function of stimulus frequency and intensity; different panels show responses from different electrodes), whereas in figure 3 we show spatial tuning curves (response strength as a function of electrode number and stimulus intensity, different panels show different stimuli). Even though these plots look similar, they are not the same and we would thus prefer to not change their orientation.

3. Fig. 3: labeling of y axes appears incorrect. electrode “2” should be “4”.

thanks for catching this, corrected.

4. In the main text the authors refer to figure S10 which is missing in the supplement.

We apologize for the missing figure. We have now added this figure as supplemental figure S13 to the revised version of the manuscript.

5. Figure S1: In the legend the authors refer to “lowest” and “highest” slopes. Slopes are steep or shallow, but not high or low.

Corrected.

6. Figure S4: Text is missing in the figure legend.

thanks for catching this, we have now added the supplemental figure legends in the correct way.

7. Figures S5 & S7: Other than indicated in the legend, the mean values are not plotted.

thanks for catching this, markers for mean values have now been added.

8. Discussion. The authors argue that red-shifted opsins will allow for more precise stimulation due to “less tissue-scattering of longer wavelength light”. The opposite is true. Scattering and, more importantly, absorption of short-wavelength light lead to a steeper fall-off of light intensity with increasing distance from the light source. Thus, blue light is more restricted in space and does not penetrate as deeply as red light. Hence, activation of opsins is more spatially confined with blue-shifted light (see e.g. Stujenske 2015).

In response to the reviewer’s comment, we have removed this statement from the manuscript.

Reviewers' Comments:

Reviewer #1:

Remarks to the Author:

This manuscript has been improved by revision. The authors have added a comparison of spread of excitation between the present gerbil study and one by Middlebrooks and Snyder in cats. The M&S study has the advantages that it includes electrical stimulation with a penetrating electrode and uses a d' analysis similar to the present study. Nevertheless, the authors might also consider looking at the studies in guinea pigs by Snyder et al. (JARO 5:305-322, 2004) and Snyder et al. (Hearing Res 235: 23-38, 2008). The reasons are: (1) that the guinea pig cochlea is smaller than that of cat, closer to that of the gerbil; and (2) the physical radii of the spiral of the gerbil and guinea pig cochleae are narrower than that of the cat (i.e., the gerbil and guinea pig cochleae are more tightly wrapped), which might tend to increase tonotopic spread of electrical current.

A few minor comments:

Line 30: a comma is missing in "electric, and"

Lines 55-56, "Using infrared stimulation it has been suggested": These are actual experimental observations that are cited, not "suggestions". One could say something like "Studies of cochlear activation using infrared stimulation have demonstrated..."

Line 60: a comma is missing in "debate, and"

Lines 242-244: This sentence doesn't make sense. Something is missing in the clause after the comma.

Lines 280-281: It would be helpful to point out that bipolar stimulation was commonly used in earlier clinical applications, but the large majority of present-day devices use monopolar stimulation.

John Middlebrooks, UC Irvine

Reviewer #2:

Remarks to the Author:

The revised version of the manuscript adequately addresses all previous concerns. The inclusion of bipolar stimulation now demonstrates much more convincingly the advantage of optogenetic stimulation over electrical stimulation with regard to spectral selectivity. Also, the more thorough discussion and the additional controls make the manuscript more convincing for a broad, non-expert readership.

Especially the fact that bipolar stimulation is not clearly superior to monopolar stimulation in the relatively small cochlea of gerbil, while optogenetic stimulation outperforms both methods, makes a strong point for the latter. With respect to clinical application, improving already well-established electrical stimulation strategies might present an easier way forward, but the much better spectral selectivity reached with optogenetic stimulation further justifies pursuing this strategy as an alternative route towards hearing restoration.

Obviously, potential caveats remain with respect to future clinical applications. The delivery and long-term expression of opsins in human tissue still remains challenging and cytotoxic side effects both of chronic light delivery and long-term overexpression of opsins need better characterization in future studies.

However, the current manuscript, as is, marks an important stepping stone on the way to

employing optogenetics as prosthetics for sensory organs.

There are still some minor issues with some figures:

Figure 2: The means mentioned in the legend are missing in panels I-K

Figure 4: The means mentioned in the legend are missing in panels C-E

Figure 4: Panel J is referred to as H in the legend

Responses to reviewer comments of the manuscript

“Near physiological spectral selectivity of cochlear optogenetics”

Alexander Dieter, Carlos J. Duque-Afonso, Dr. Vladan Rankovic, Dr. Marcus Jeschke,
Prof. Tobias Moser

Reviewer #1 (Remarks to the Author):

This manuscript has been improved by revision. The authors have added a comparison of spread of excitation between the present gerbil study and one by Middlebrooks and Snyder in cats. The M&S study has the advantages that it includes electrical stimulation with a penetrating electrode and uses a d' analysis similar to the present study. Nevertheless, the authors might also consider looking at the studies in guinea pigs by Snyder et al. (JARO 5:305-322, 2004) and Snyder et al. (Hearing Res 235: 23-38, 2008). The reason are: (1) that the guinea pig cochlea is smaller than that of cat, closer to that of the gerbil; and (2) the physical radii of the spiral of the gerbil and guinea pig cochleae are narrower than that of the cat (i.e., the gerbil and guinea pig cochleae are more tightly wrapped), which might tend to increase tonotopic spread of electrical current.

We are grateful to Prof. Middlebrooks for the appreciation of our work and his comments that helped us to improve our manuscript. As suggested, we have added a comparison to electrical stimulation of the guinea pig cochlea in the discussion section.

A few minor comments:

Line 30: a comma is missing in “electric, and”
Done.

Lines 55-56, “Using infrared stimulation it has been suggested”: These are actual experimental observations that are cited, not “suggestions”. One could say something like “Studies of cochlear activation using infrared stimulation have demonstrated...”

We have adapted this sentence.

Line 60: a comma is missing in “debate, and”
Done.

Lines 242-244: This sentence doesn't make sense. Something is missing in the clause after the comma.

Rephrased for clarity.

Lines 280-281: It would be helpful to point out that bipolar stimulation was commonly used in earlier clinical applications, but the large majority of present-day devices use monopolar stimulation.

We are grateful for pointing this out and have included this statement.

John Middlebrooks, UC Irvine

Reviewer #2 (Remarks to the Author):

The revised version of the manuscript adequately addresses all previous concerns. The inclusion of bipolar stimulation now demonstrates much more convincingly the advantage of optogenetic stimulation over electrical stimulation with regard to spectral selectivity. Also, the more thorough discussion and the additional controls make the manuscript more convincing for a broad, non-expert readership.

Especially the fact that bipolar stimulation is not clearly superior to monopolar stimulation in the relatively small cochlea of gerbil, while optogenetic stimulation outperforms both methods, makes a strong point for the latter. With respect to clinical application, improving already well-established electrical stimulation strategies might present an easier way forward, but the much better spectral selectivity reached with optogenetic stimulation further justifies pursuing this strategy as an alternative route towards hearing restoration.

Obviously, potential caveats remain with respect to future clinical applications. The delivery and long-term expression of opsins in human tissue still remains challenging and cytotoxic side effects both of chronic light delivery and long-term overexpression of opsins need better characterization in future studies. However, the current manuscript, as is, marks an important stepping stone on the way to employing optogenetics as prosthetics for sensory organs.

We thank Professor Wiegert for his previous comments that helped us to improve our manuscript, as well as for the appreciation of the work we did during the review process. We are glad that we were able to address all of his the concerns.

There are still some minor issues with some figures:

Figure 2: The means mentioned in the legend are missing in panels I-K
Markers for mean values have been added.

Figure 4: The means mentioned in the legend are missing in panels C-E
Markers for panels C, D and E have been added.

Figure 4: Panel J is referred to as H in the legend
Corrected.

Simon Wiegert